# Hard-Templated Porous Niobia Films for Optical Sensing Applications

Venelin Pavlov, Rosen Georgiev *, Katerina Lazarova , Biliana Georgieva and Tsvetanka Babeva *

Institute of Optical Materials and Technologies "Acad. J. Malinowski", Bulgarian Academy of Sciences, Akad. G. Bonchev Str., Bl. 109, 1113 Sofia, Bulgaria
* Correspondence: rgeorgiev@iomt.bas.bg (R.G.); babeva@iomt.bas.bg (T.B.)

**Abstract:** Porous $Nb_2O_5$ films obtained by a modified hard-template method were studied and their optical and sensing properties were optimized in order to find applications in chemo-optical sensing. Porous films were prepared by following three steps: liquid mixing of niobium sol and $SiO_2$ colloids in different volume fractions, thermal annealing of spin-coated films for formation of a rigid niobia matrix, and selective removal of silica phase by wet etching thus generating free volume in the films. The morphology and structure of the films were studied using transmission electron microscopy and selected area electron diffraction, while their optical and sensing properties were estimated using UV-VIS-NIR reflectance measurements in different ambiences such as air, argon and acetone vapors and nonlinear curve fitting of the measured reflectance spectra. Bruggeman effective medium approximation was applied for determination of the volume fraction of silica and air in the films, thus revealing the formation of porosity inside the films. For further characterization of composite films, their water contact angles were measured and finally conclusions about the impact of initial chemical composition and etching duration on properties of the films were drawn.

**Keywords:** $Nb_2O_5$; hard-template method; Ludox®; porous materials; etching

## 1. Introduction

The increasing demand for sensing applications in health and environmental monitoring, food quality control, and the automotive and aerospace industries has lead to significant interest in the development of reliable and sensitive technology that can meet many industries' requirements. Optical sensing is a powerful tool for the detection of different chemical agents [1,2]. It is immediate, sensitive, and relatively easy to implement. The detection can be performed by the naked eye and does not require a power source, which are great advantages when compared with other technologies, and stands out as a promising method for even broader future implementation.

Porous media have long been proposed as a platform for optical sensing due to their properties—enhanced surface area, pore size tunability, simple functionalization strategies with organics, biocompatibility, low toxicity, and their ability to interact with different chemical agents. The common materials reported include carbons [3], MOFs [4], zeolites [5], and different transitional metal oxides [6–8]. The variety of applications, however, requires a broad range of material properties and methods of synthesis which leads to an ever-increasing demand for tailoring them.

One of the main benefits of utilizing transitional metal oxides for optical sensing is their transparency in the visible spectrum. Often, thin films are integrated in Bragg stacks in order to improve the optical response and wavelength selectivity of the sensing element [9]. More often than not, the sensing medium in Bragg stacks are polymers [10], which are not as durable as inorganic materials. That is why many porous transitional metal oxides are proposed as sensing elements. However, little attention has been given to $Nb_2O_5$.

Niobium oxide is a transitional metal oxide with great potential for numerous applications [11]. Due to its crystal structural variety, electronic bandgap, and optical and mechanical properties it is a promising candidate for implementation in different realms. $Nb_2O_5$-based MIM capacitors were investigated in [12], and in [13] their mechanical properties were tested on a flexible substrate. A thin film of this material was successfully demonstrated as a wide band anode material for dye-synthesized solar cells [14] and double junction solar cells [15]. Due to their chemical stability they are often proposed as antireflection coatings [16,17] and corrosion barrier coatings [18].

Recently we have demonstrated preparation of porous $Nb_2O_5$ films using a soft template method where block copolymers (commercially available Pluronic PE polymers) were used as a template for generation of porosity [19]. The method benefits from low cost and the easy synthesis pathway of different pore shapes and dimensions, depending on the experimental conditions [20]. In order to obtain films with stronger porosity, a higher amount of polymer template has to be used. However, more polymer in the film leads to thinner walls of the oxide skeleton, and after annealing even at moderate temperatures a collapse is very likely to take place. An alternative approach for the preparation of porous materials is the hard-template method, also known as nanocasting, where a solid template with well defined porosity at the nanometer scale is infiltrated with the target material and after removal of the template a functional structure with inverse morphology is obtained [21–23]. The main difficulties related to hard-templating are the time consuming processes of template preparation, infiltration of the targeted material inside the template due to low mass transfer at the nanoscale [24], and removal of templates that could destroy the porous materials. However, the method is linked with extremely uniform pore size distribution [24].

To overcome the aforementioned drawbacks, in this paper, we utilize a modified hard-template method for preparation of thin $Nb_2O_5$ films with tunable porosity and study their suitability as active media for VOC sensing. The films are prepared using silica colloids (Ludox®) that are added to niobium sol and spin-coated on silicon substrates. In order to tailor the porosity, different volume ratios of Ludox® to Nb sol are used. Thereafter the template is removed by selective chemical etching and the porous medium is characterized. The refractive index modulation and thickness of the composite films are determined by nonlinear curve fitting method. The sensing properties of the films are tested with acetone vapors as a representative of VOCs. The potential of using the studied mesoporous $Nb_2O_5$ films for chemo-optical sensing is demonstrated and discussed.

## 2. Materials and Methods

The common preparation of hard-templated porous films consists of deposition of a hard template, subsequent infiltration of the target material, and selective removal of the template. However, in our study we adopted a different approach. Instead of preparation of the hard template and infiltration with niobia, we used a liquid mixture of Nb sol and a colloidal solution of $SiO_2$ nanoparticles. The commercially available product Ludox® (LUDOX® AS-30 colloidal silica, Grace, Worms, Germany) was utilized as a hard template. It is a stabilized colloidal solution of silica spheres with nominal particle size of 12 nm and weight concentration of 30 wt%. The $SiO_2$ template was diluted with distilled water in a ratio of 1:1 and added to the already prepared niobium sol in different volume ratios: Nb sol-to- Ludox® = 50:1, 20:1, 10:1 and 5:1. Niobium sol was prepared using the sonocatalytic method using 0.400 g NbCl5 (99%, Sigma-Aldrich, Saint Louis, MO, USA) as a precursor and 8.3 mL ethanol (96%, Chempur, Piekary Śląskie, Poland) mixed with 0.17 mL distilled water as a solvent, according to the methods described in detail in [25]. The solution aged for 24 h at ambient conditions prior to mixing with the template.

Thin composite films were obtained by spin-coating (Laurell WS-650MZ-23NPPB spin coater, North Wales, PA, USA) at a rotation rate of 4000 rpm and an acceleration rate of 1500 rpm/s, and thermally treated at 320 °C for 30 min (increasing temperature rate of 5 °C/min) in order for the niobia matrix to be created. For the selective removal of silica,

composite thin films were immersed in a mixture of 0.3 mL nitric acid (Chempur-Poland), 0.5 mL hydrofluoric acid (Chempur-Poland), and 109.2 mL distilled water for different durations and rinsed with deionized water. It was found that at higher acid concentrations films cracked during the etching process. In such a way, the optimal concentration at which no cracks appeared in the films was achieved.

The morphology and structure of the films were studied using Transmission Electron Microscopy and Selected Area Electron Diffraction (SAED), respectively, using a high resolution transmission electron microscope JEOL JEM 2100 (JEOL Ltd., Tokyo, Japan).

Optical and sensing properties of the films were investigated using reflectance measurements in different ambient surroundings—air, argon, and acetone vapors—using a UV-VIS-NIR spectrophotometer Cary 05E (Varian, Australia). Refractive index and thickness were calculated using nonlinear curve fitting of reflectance spectra measured in air. Sensing behavior was estimated by using refractive index change calculated from the change of reflectance spectra of the samples due to exposure from argon and acetone vapors.

Wetting behavior of the composite films was studied using a DSA30 Drop Shape Analyzer (KRÜSS GmbH, Hamburg, Germany) via subsequently measuring their respective static water contact angles (WCAs). Measurements of WCAs were taken after immersing the samples in the standard etcher for different times (varying from 0 to 240 s), rinsing with DI water and spinning at 4000 rpm for 30 s for drying. Approximately 8 μL of water was dispensed on the samples' surface and multiple readings (>30) of WCAs were taken 1–2 s after that. Measurements were performed at room temperature and RH of 55%. In addition, measurements of WCAs of pre-cleaned bare substrate, pure niobia, and Ludox® films were conducted and compared with values for the composite films.

## 3. Results and Discussions

### 3.1. TEM Characterization of Thin Composite Films

The first step of our investigation was the characterization of the shape, size and structure of the nanoparticles. Figure 1 shows a TEM image of $SiO_2$ nanoparticles and their size distribution. The particles had a spherical shape, quite uniform distribution and average size of 13 nm. The particles were well separated from each other and no aggregation was observed. Selected area electron diffraction (SAED) (not shown in Figure 1) proved the amorphous status of the silica particles.

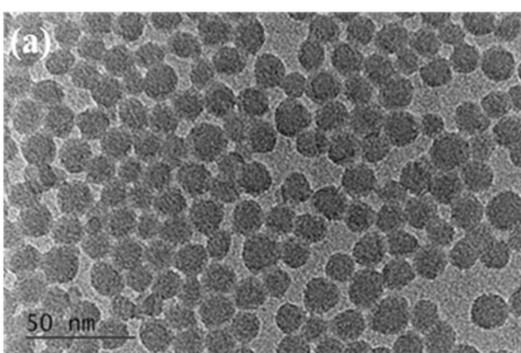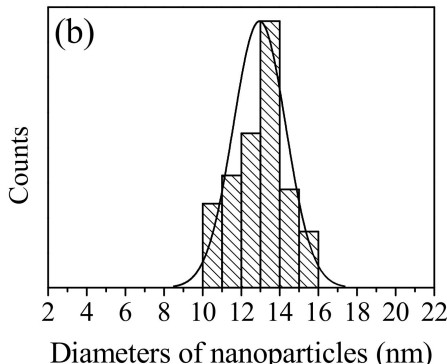

**Figure 1.** TEM pictures of $SiO_2$ particles in Ludox® (**a**) and histogram distribution of their size (**b**).

Once the template had been characterized, we continued with the investigation of its influence on the morphology of the $Nb_2O_5$ thin films. Figure 2 presents TEM pictures and a typical SAED pattern of hard-templated $Nb_2O_5$ films prepared with different Nb sol to Ludox® ratios after annealing at 320 °C.

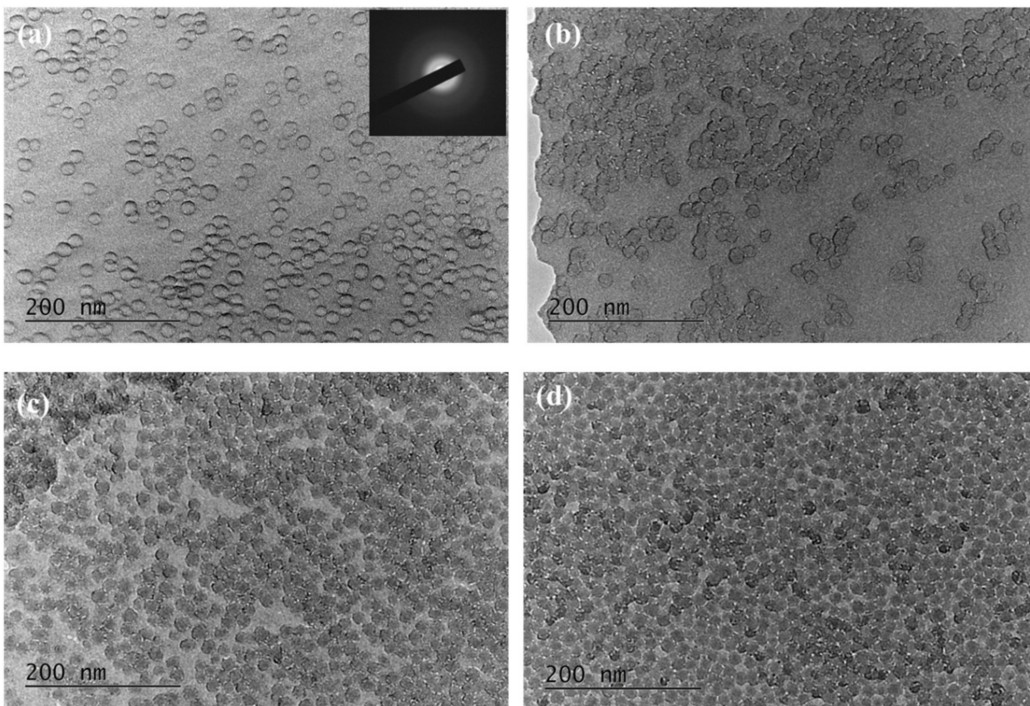

**Figure 2.** TEM images of porous films of $Nb_2O_5$ prepared by mixing Nb sol and a colloidal solution of nanoparticles of $SiO_2$ (commercially available LUDOX®) in different volume fractions: 50:1 (**a**), 20:1 (**b**), 10:1 (**c**) and 5:1 (**d**); Typical selected area electron diffraction (SAED) pattern is shown as an inset.

From the images in Figure 2 it can be seen that increasing the amount of nanoparticles in the sol increases the volume fraction of the particles inside the $Nb_2O_5$ film. In all four cases under investigation, the particles were distributed evenly over the entire surface of the films, and no aggregated clusters were observed. The SAED patterns were taken of each film, but for the sake of brevity only for the first one is shown. They confirm the amorphous nature of the films. The comparison of Figures 1 and 2 shows that after the films annealing, $SiO_2$ nanoparticles inside the composite films preserve their size and shape. This was expected, considering that $SiO_2$ nanoparticles are temperature-stable and durable.

### 3.2. Optical Characterization and Generation of Porosity

Wavelength dependence of refractive index (the so-called refractive index dispersion) of composite films are presented in Figure 3. It is seen from Figure 3 that the volume ratio of the two phases impacts substantially refractive index values, *n* of the films. As $SiO_2$ increases, *n* decreases from 1.80 for 50:1 sample to 1.65, 1.48 and 1.26 for 20:1, 10:1 and 5:1 samples, respectively at a wavelength of 600 nm. Considering that the refractive index of $SiO_2$ (1.46 at 600 nm [26]) is smaller than that of $Nb_2O_5$ (2.1 at 600 nm [19]) it could be expected the effective refractive index of the composite to decrease when the volume fraction of silica increases. For completeness we have deposited films consisting only of $SiO_2$ nanoparticles and calculated their refractive index to be 1.25 at 600 nm, which is very close to the refractive index of 5:1 sample.

The next step of our investigation was to generate porosity in the studied films through selective removal of the silica phase. Different levels of porosity were accomplished in the niobium pentoxide layers with added silicon oxide nanoparticles by etching the $SiO_2$ phase by immersing the films for different times (from 0 to 240 s) in a standard etching solution. The etching rates of $Nb_2O_5$ and $SiO_2$ are very different, enabling in this way the selective removal of the $SiO_2$ phase. For this purpose, layers with different volume ratios of Nb sol to template were deposited and their reflectance spectra were measured after each etching step. Results are depicted in Figure 4.

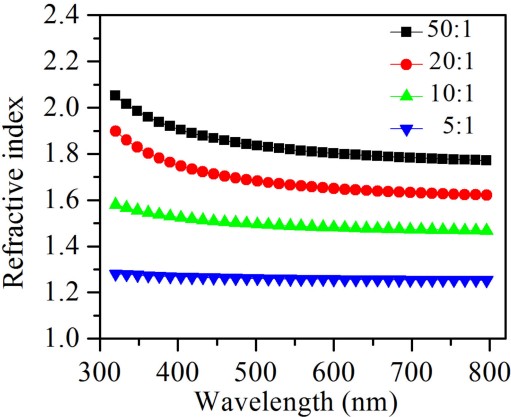

**Figure 3.** Dispersion of refractive index of $Nb_2O_5$ films prepared by mixing of Nb sol and colloidal solution of nanoparticles of $SiO_2$ in different volume fractions indicated in the figure.

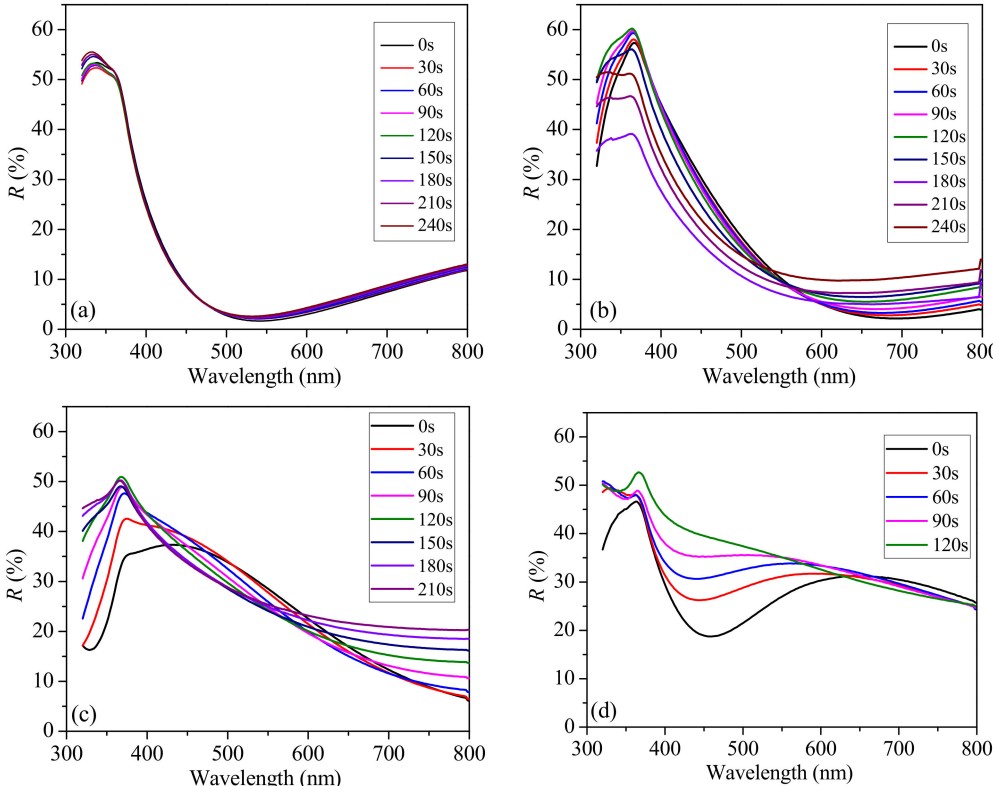

**Figure 4.** Reflectance spectra of hard-templated $Nb_2O_5$ obtained by liquid-phase mixing of Nb sol and silica colloids in volume ratios of 50:1 (**a**), 20:1 (**b**), 10:1 (**c**) and 5:1 (**d**) during selective removal of $SiO_2$ phase by wet-etching.

It can be seen from the figure that when the $Nb_2O_5$ phase prevailed (ratio 50:1, Figure 4a), etching took place at a very slow rate—the change in the reflectance spectra due to etching up to 240 s is very weak. On the contrary, when the $SiO_2$ phase prevailed (ratio 5:1, Figure 4d) the etching was much more effective—a substantial change in the reflectance spectrum is seen at each etching step. Spectra of 10:1 and 20:1 samples changed continuously during etching.

To gain more comprehensive insights into the final composite formation, the measured reflectance spectra were used to determine the thickness and refractive index of the films after each etching step. The results for the refractive index and thickness are presented in Figure 5 and Table 1, respectively.

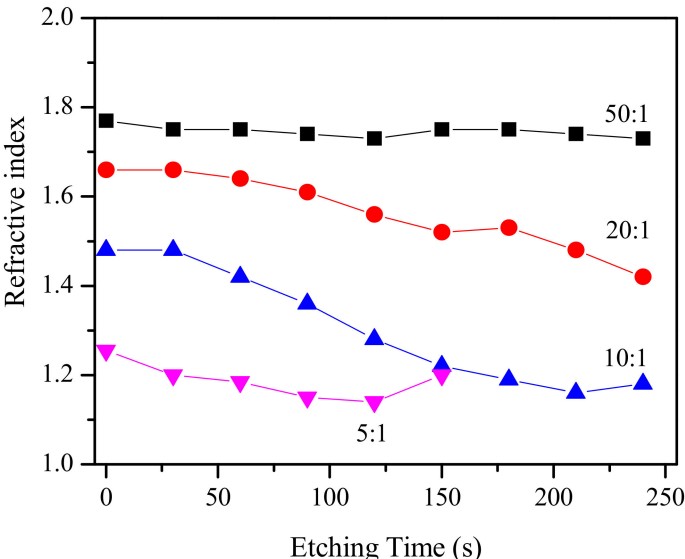

**Figure 5.** Refractive index change as a function of etching time of porous $Nb_2O_5$ layers obtained by liquid phase mixing of niobium sol and colloidal solution of $SiO_2$ nanoparticles in different volume ratios indicated in the figure.

**Table 1.** Dependence of film thickness, $d$ (nm), and volume fractions of niobia, $f_{niobia}$, silica, $f_{silica}$, and air, $f_{air}$, in hard-templated niobia films after each etching step.

| Etching Time (s) | 50:1 | | | | 20:1 | | | | 10:1 | | | | 5:1 | | | |
|---|---|---|---|---|---|---|---|---|---|---|---|---|---|---|---|---|
| | $d$ (nm) | $f_{niobia}$ (%) | $f_{silica}$ (%) | $f_{air}$ (%) | $d$ (nm) | $f_{niobia}$ (%) | $f_{silica}$ (%) | $f_{air}$ (%) | $d$ (nm) | $f_{niobia}$ (%) | $f_{silica}$ (%) | $f_{air}$ (%) | $d$ (nm) | $f_{niobia}$ (%) | $f_{silica}$ (%) | $f_{air}$ (%) |
| 0 | 75 | 65 | 14 | 21 | 102 | 46 | 34 | 20 | 156 | 29 | 38 | 33 | 266 | 16 | 23 | 61 |
| 30 | 76 | 65 | 10 | 25 | 102 | 46 | 34 | 20 | 151 | 29 | 38 | 33 | 259 | 16 | 10 | 74 |
| 60 | 76 | 65 | 10 | 25 | 102 | 46 | 30 | 24 | 150 | 29 | 26 | 45 | 248 | 16 | 5 | 79 |
| 90 | 76 | 65 | 8 | 27 | 102 | 46 | 24 | 30 | 154 | 29 | 15 | 56 | 239 | 16 | 0 | 84 |
| 120 | 76 | 65 | 6 | 29 | 103 | 46 | 14 | 40 | 155 | 29 | 0 | 71 | 220 | 16 | 0 | 84 |
| 150 | 76 | 65 | 10 | 25 | 105 | 46 | 8 | 46 | 158 | 24 | 0 | 76 | 100 | 21 | 0 | 79 |
| 180 | 75 | 65 | 10 | 25 | 105 | 46 | 12 | 42 | 159 | 21 | 0 | 79 | - | - | - | - |
| 210 | 75 | 65 | 9 | 27 | 104 | 46 | 0 | 54 | 158 | 18 | 0 | 82 | - | - | - | - |
| 240 | 76 | 65 | 6 | 29 | 107 | 37 | 0 | 63 | 124 | 20 | 0 | 80 | - | - | - | - |

Figure 5 demonstrates a continuous decrease in the refractive index with etching time for all studied films. However, the reduction rate of $n$ is different and depends on the initial chemical composition of the films. The strongest change in refractive index was observed for films with sol to template ratios of 20:1 and 10:1. It is interesting to note that at up to 240 s for the first film and up to 210 s for the second there was no thickness change (Table 1). For 50:1 films where $Nb_2O_5$ prevailed, the change in the refractive index was very weak and no thickness change was observed for etching up to 240 s. On the contrary, for films with a predominant $SiO_2$ phase, in addition to a decrease in the refractive index a monotonous decrease in thickness took place that continued up to 120 s, and thickness changed from 266 nm to 220 nm. A further 30 s etching led to a strong dissolution of the films as indicated by a sharp decrease in the thickness from 220 nm to 100 nm.

To clarify further the process of selective etching we calculated the volume fractions $f_{niobia}$ and $f_{silica}$ of both solid phases, silica and niobia, and the free volume in the film $f_{air}$ (i.e., the volume fraction of air). The latter was assumed to exist considering the very low refractive index values of 5:1 films that were even lower than the refractive index of $SiO_2$. It is clear that the effective refractive index of a medium comprising two constituents with approximate $n$-values of 2.1 and 1.46 can be less than 1.46 only if air with a refractive index of the unit is present inside.

Thus the composite film is regarded as an effective medium with dielectric constant $\varepsilon_e$ comprising three phases: niobia, silica, and air, and is described by the Bruggeman effective medium approximation:

$$f_{niobia}\frac{\varepsilon_n - \varepsilon_e}{\varepsilon_n + 2\varepsilon_e} + f_{silica}\frac{\varepsilon_s - \varepsilon_e}{\varepsilon_s + 2\varepsilon_e} + f_{air}\frac{1 - \varepsilon_e}{1 + 2\varepsilon_e} = 0 \qquad (1)$$

$$f_{niobia} + f_{silica} + f_{air} = 1 \qquad (2)$$

where $\varepsilon_n$ and $\varepsilon_s$ are the dielectric constants of dense niobia and silica, respectively, and $f_{niobia}$ and $f_{silica}$ are their volume fractions. $f_{air}$ is the volume fraction of air with dielectric constants of 1. For clarity, it is important to note that the two parameters $\varepsilon$ and $n$ are related as follows: $\varepsilon = n^2$.

The volume fraction of niobia was calculated as the ratio of thickness of pure $Nb_2O_5$ film (51 nm) to the thickness of composite films. The dilution of Nb sol due to the addition of colloidal silica solution was taken into account and the thickness of pure niobia film was reduced accordingly to 49 nm, 47 nm, 44 nm and 43 nm for 50:1, 20:1, 10:1 and 5:1 films, respectively. These values of thickness are taken from our previous results concerning a calibration curve of $Nb_2O_5$ film thickness as a function of Nb sol concentration [25].

The volume fractions of niobia and silica are calculated from Equations (1) and (2) and all results are included in Table 1. It is seen that during etching the volume fraction of silica decreased and the free volume in the films increased. For the 50:1 film an etching duration of 240 s does not lead to complete removal of $SiO_2$, while for the rest of the films even shorter time is sufficient to utterly dissolve silica: 210 sec, 120 s and 90 s for 20:1, 10:1 and 5:1 films, respectively. It is interesting to note that the free volume in the 20:1 and 10:1 films continued to increase after complete removal of $SiO_2$: $f_{air}$ increased from 54% to 63% for the 20:1 film and from 71% to 82% for the 10:1 film. This means that a slight dissolution of the niobia phase takes place, which generates additional porosity in the films. Further etching led to the collapse of the niobia skeleton that is accompanied with a decrease in thickness and free volume fraction most pronounced for the 5:1 films.

### 3.3. Wetting Behavior of Composite Films

The next step of our investigation was to study the wetting behavior of the composite films by measuring their static water contact angles (WCAs). Since it is known from the literature that niobia is hydrophobic and silica is hydrophilic, it was assumed that by measuring WCAs of composite films at each etching step it would be possible to conclude about the ratio between the two phases in the films and obtain more knowledge about the nature of porosity formation.

In the first step, the water contact angles of the films were measured before etching and the results are presented in Figure 6. It is seen that as the $SiO_2$ phase increased, the contact angle decreased from 58° at a sol to template ratio of 50:1 to 14° at 5:1. The measured contact angles of $Nb_2O_5$ and $SiO_2$ layers are 59° and 20°, respectively, which confirms the assumption that $Nb_2O_5$ is much more hydrophobic than $SiO_2$. Therefore, when the hydrophilic phase of $SiO_2$ was added to the hydrophobic $Nb_2O_5$, the resulting effective medium became more hydrophilic, i.e., the contact angle decreased.

The influence of etching duration on water contact angles of composite films are depicted in Figure 7. It can be seen from Figure 7 that films with a predominant $SiO_2$ phase (ratio 5:1) became more hydrophobic with etching (WCA increased from 14° to 38°). This can be caused by dissolving the more hydrophilic phase in the effective medium, thus increasing the volume fraction of the more hydrophobic phase. For films where the $Nb_2O_5$ phase prevailed, no significant change in water contact angle took place with etching. The contact angle values for these layers varied around that of pure $Nb_2O_5$.

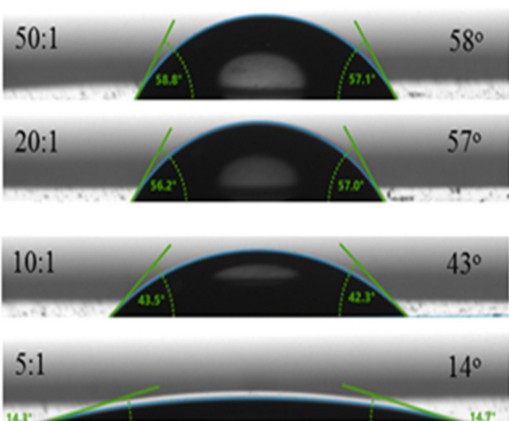

**Figure 6.** Water contact angles of hard-templated $Nb_2O_5$ films obtained by liquid phase mixing of Nb sol and silica colloids in different volume ratios indicated in the figure.

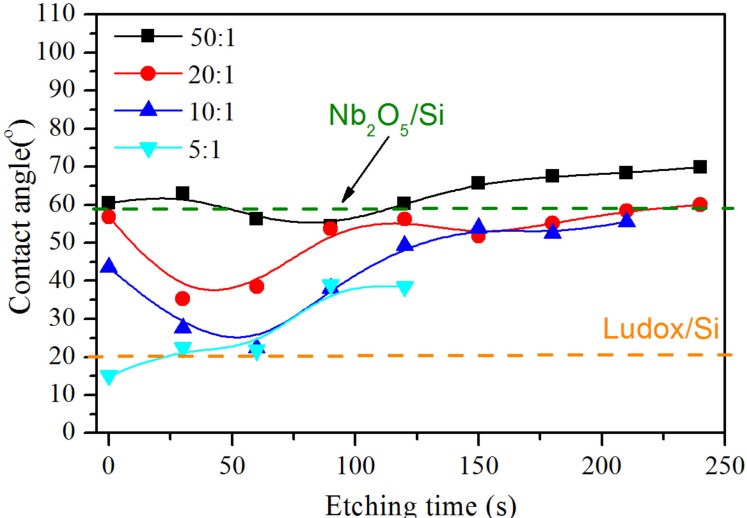

**Figure 7.** Water contact angles as a function of etching time of hard-templated porous $Nb_2O_5$ films obtained by liquid phase mixing of Nb sol and silica colloids in different volume ratios indicated in the figure. Water contact angles of pure Ludox® and niobia films are represented by the two dashed horizontal lines.

For films with 20:1 and 10:1 sol to template ratios, the same trend of decreasing contact angle was observed in the initial stage of etching, after which the values increased again and reached those of pure $Nb_2O_5$. It is possible that in the beginning of the dissolution process, the nitric acid from the etching solution promoted the formation of hydroxyl groups on the surface, resulting in enhancement of wetting, i.e., a decrease in WCAs. As the etching time increased, the hydrophilic phase in the layer dissolved and the water contact angle increased due to the prevailing of the $Nb_2O_5$ phase. The calculated volume fraction of silica confirmed that for 20:1 and 10:1 films the silica phase substantially decreased after 90 s of etching (Table 1) and this is the possible reason for the observed increase in WCAs.

### 3.4. Sensing Response toward Acetone Vapors

The final step of our investigation concerned the sensing behavior of the studied films. Our previous sensing experiments have shown that pure sol-gel $Nb_2O_5$ films do not change their refractive index when exposed to vapors of volatile organic compounds. If we assume that silica nanoparticles in the studied hard-templated $Nb_2O_5$ films are tightly surrounded by the niobia matrix and there are no free spaces inside the films, it could be expected that films do not change their refractive index when exposed to vapors. However,

measurements have demonstrated a change in refractive index of the films before removal of silica that increased with nanoparticle content in the films (Figure 8). These observations are in line with the results of calculated volume fractions of air in the films that confirm the increase in free volume with the addition of colloidal nanoparticles.

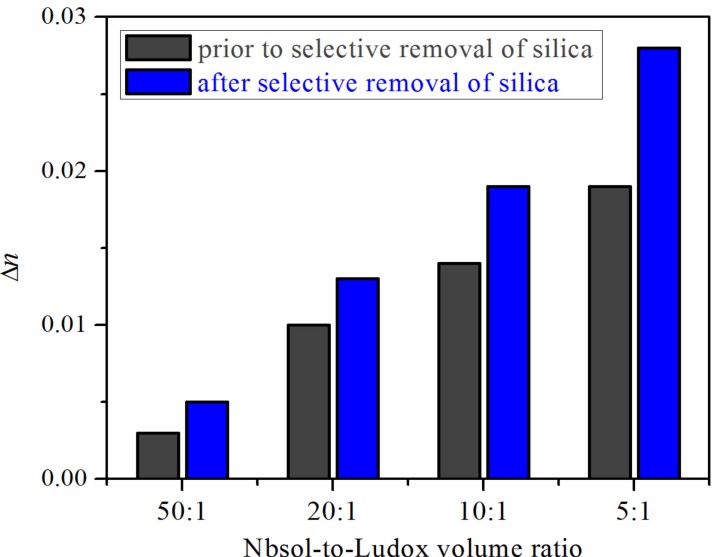

**Figure 8.** Refractive index change due to acetone vapor absorption prior to and after selective removal of silica in hard-templated films of $Nb_2O_5$ obtained by liquid phase mixing of Nb sol and silica colloids in different volume ratios indicated in the figure. The films were etched in a standard $SiO_2$ etcher for different times: 240 s for the 50:1 and 20:1 films, 210 s for the 10:1 film and 120 s for the 5:1 film.

After etching, an enhancement of the sensor response was observed for all films that confirms the results for volume fraction showing an increase in the free volume in the films with etching. It is concluded that etching increases the porosity in the films [27] thus facilitating the adsorption and condensation of vapors in the generated pores, which finally leads to an increase in the effective refractive index of the films. The sensing experiments in Figure 8 are conducted at an etching duration of 240 s for the 50:1 and 20:1 films, 210 s for the 10:1 films and 120 s for the 5:1 films.

In order to optimize the sensor response further, the etching duration was varied in the range from 0 to 210 s for the 10:1 sample and from 0 to 240 s for the 20:1 sample. Reflectance spectra were measured in an ambient of argon and acetone vapors, and the change in refractive index $\Delta n$ was calculated from the reflectance change. The results are presented in Figure 9. It is seen from Figure 9 that for both samples there is an optimal etching duration that leads to maximal refractive index change and it depends on the initial amount of silica in the films. When this amount is less (sample 20:1), the optimum value is around 150 s and shifts to 100 s with increasing amount of $SiO_2$ phase in the composite films (10:1 sample). If we look again at the results in Table 1 we see that the best sensing performance of both films is for etching durations that allow small amounts of silica to remain in the films.

One factor that governs the analyte adsorption ability is the chemical affinity of the analyte to the constituent layers. M. N. Ghazzal et al. have studied a $TiO_2/SiO_2$ multilayer system and have demonstrated that tuning the hydrophobic–hydrophilic balance of the silica surface modifies the affinity of the mesoporous multilayer system to the environment [28]. Similarly we may assume here that the presence of a small amount of silica increases the chemical affinity to acetone vapors, thus enhancing adsorption processes.

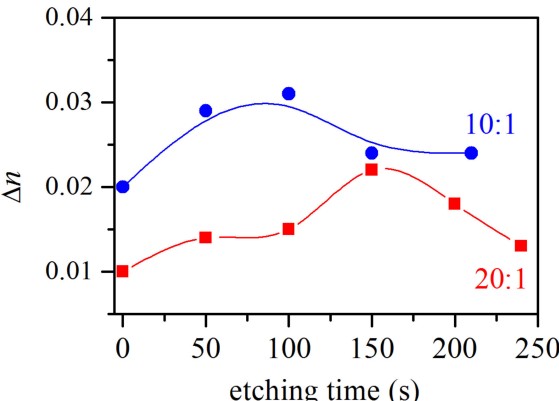

**Figure 9.** Refractive index change as a function of etching time of hard-templated films of Nb$_2$O$_5$ obtained by liquid phase mixing of Nb sol and silica colloids in different volume ratios indicated in the figure.

## 4. Conclusions

A modified hard-template method was utilized and the successful preparation of vapor sensitive Nb$_2$O$_5$ thin films with tailored porosity and tunable optical properties was demonstrated. The preparation method comprised three main steps: mixing the appropriate amount of Nb sol and silica colloids (commercially available Ludox®), thermal treatment of the spin-coated composite films, and selective etching of the silica phase. It was shown that morphology, optical and sensing properties, and wetting behavior depend both on the initial chemical composition of the films and the etching duration. An optimal etching time was found that maximizes the sensing response of the films towards acetone vapors. It was revealed that the best sensing performance of the films was achieved when a small amount of silica remained in the films.

**Author Contributions:** Conceptualization, R.G. and T.B.; methodology, R.G. and T.B.; software, V.P., R.G. and K.L.; validation, R.G. and T.B.; formal analysis, V.P., R.G. and K.L.; investigation, V.P., B.G. and K.L.; resources, R.G.; data curation, V.P. and K.L.; writing—original draft preparation, R.G., K.L. and T.B.; writing—review and editing, R.G. and T.B.; visualization, T.B. and R.G.; supervision, R.G. and T.B.; project administration, R.G.; funding acquisition, R.G. All authors have read and agreed to the published version of the manuscript.

**Funding:** This research was funded by the Bulgarian National Science Fund, Grant No. KP06-M48/3 (26 November 2020).

**Institutional Review Board Statement:** Not applicable.

**Informed Consent Statement:** Not applicable.

**Data Availability Statement:** Not applicable.

**Acknowledgments:** Research equipment of INFRAMAT (part of Bulgarian National roadmap for research infrastructures) supported by Bulgarian Ministry of Education and Science was used in this investigation. T.B. acknowledges the partial support of CoE "National center of mechatronics and clean technologies" BG05M2OP001-1.001-0008-C01 supported by the European Regional Development Fund within the Operational Programme "Science and Education for Smart Growth 2014–2020".

**Conflicts of Interest:** The authors declare no conflict of interest.

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
