# Peer review of "Hard-Templated Porous Niobia Films for Optical Sensing Applications"

_photonics, doi:10.3390/photonics10020167_

Round 1

Reviewer 1 Report

This manuscript contributed the synthesis of porous structural Nb2O5 film using hard-templating method with commercially available 12 nm SiO2 colloids (LUDOX® AS-30 colloidal silica, Merck) as templates and NbCl5 (99%, Aldrich) as an inorganic precursor. In addition, spin-coating technique was used for film fabrication process, and a porous film was obtained after high-temperature calcination and acid etching for pore exposing. Using transition electron microscopy, selected area electron diffraction, and UV-

VIS-NIR reflectance measurements, it likely shows the formation of porous structure film with its application as chemooptical sensing toward VOC’s like acetone vapor. However, this manuscript cannot be recommended for publication at this shape. Here are some suggestions:

1) For porous film preparation. Here should be more details about the hard-template synthesis process. What’s the interaction between SiO2 colloids and inorganic precursor?

How is the solvation process happening for Nb sol preparation?

Why the prepared solution needs an aging process?

2) Structural characterization. What’s the pore size of Nb2O5 film after calcination like showed in Figure 2, is that comparable to that of SiO2 colloids?

3) What’s the role of remain silica in the film, since it gives out the best performance?

4) Couple of writings over the manuscript lack relative background literature/evidence, as shown below:

Page 2 ‘hard-template method also known as nanocasting where a solid template with well-defined porosity on nanometer scale is infiltrated with target material and after removal of the template a functional structure with inverse morphology is obtained.’

Page9 ‘etching increases the porosity in the films thus facilitating the adsorption and condensation of vapors in the generated pores that finally leads to an increase in effective refractive index of the films.’

5) It necessary to discuss more about the differences or dis/advantages of soft and hard-template method, since mentioned both of them earlier.

Author Response

Dear Reviewer, thank you very much for your positive assessment of our work and for your fruitful recommendations and very helpful comments. Please find as follows our answers. Please have in mind that all your comments and recommendations are considered and the manuscript text is revised accordingly everywhere it is necessary. The changes in the text are in red color.

Q1.1: For porous film preparation. Here should be more details about the hard-template synthesis process. What’s the interaction between SiO2 colloids and inorganic precursor?

 Answer: The reviewer could not find details about the hard-template synthesis in our study, because we used commercially available colloidal silica (LUDOX® AS-30) as a hard template. LUDOX® is a stabilized aqueous dispersion of discrete nano-sized (nominal diameter of 12nm) spherical silica particles with amorphous structure and very narrow particle size distribution. For preparation of composite thin films Ludox was mixed with already prepared Nb sol. If the metal alkoxide in the Nb sol is not fully hydrolyzed it is possible some reactions to take place because Ludox is rich in silanol groups. However, we do not have some evidence for this nor additional studies were performed for confirmation.

Q1.2: How is the solvation process happening for Nb sol preparation?

Answer: In our previous paper (Lazarova et. al. Opt. Laser Technol 2014, 58, 114–118, Ref. 20) and in the references inside the processes of formation of Nb2O5 films through sol-gel method were explained in more detail. Briefly, usually two types of precursor were used: metal alkoxides (Nb ethoxide) (see ref. 1 and 16 inside) or metal salt-NbCl5 (see ref. 17, 18 and 19 inside). The sol-gel chemistry of metal alkoxides has been extensively studied and clarified (see for example S. M. Attia et. al., J. Mater. Sci. Technol., 18 (3), 2002, 211-218 and L. Hench and J. West, Chem. Rev. 90, 1990, 33-72). However, because Nb alkoxides are quite expensive and highly sensitive to moisture, procedures for obtaining Nb alkoxide from Nb salt have been developed (see for example M. A. Aegerter, Solar Energy Materials & Solar Cells 68 (2001) 401-422). The metal alkoxide was synthesized through the reaction of NbCl5 with alcohol as follows:

NbCl5 + x(R-OH) --> NbCl5-x(OR)x + xHCl, where R is an alkyl group.

After Nb alkoxide was synthesized the common processes of hydrolysis and condensation take place (see the references mentioned above).

Q1.3. Why the prepared solution needs an aging process?”

Answer: Our previous experience in sol-gel preparation of Nb2O5 thin films have shown that thin films should be deposited after 24 hours of Nb sol aging in order to obtain reproducible (in terms of refractive index, n, and thickness, d) sol-gel Nb2O5 films. Furthermore we have demonstrated that during the first week of sol aging the deposited films have similar values of refractive index with the highest value reached after 3 days of sol aging. If films are prepared using Nb sol aged more than a week a decrease in n is observed. The thickness of the film is almost constant for a period of 10 days of sol aging and after that starts monotonically to increase (Lazarova et. al. Opt. Laser Technol 2014, 58, 114–118, Ref. 20)

 Q2:“Structural characterization. What’s the pore size of Nb2O5 film after calcination like showed in Figure 2, is that comparable to that of SiO2 colloids?”

Answer: Actually, figure 2 demonstrates the TEM pictures of composite films after calcination. The comparison with figure 1 shows that after annealing SiO2 nanoparticles inside the composite films preserve their size and shape. This could be expected considering that SiO2 nanoparticles are temperature stable and durable.

Q3:“What’s the role of remain silica in the film, since it gives out the best performance?”

Answer: One factor that governs the analyte adsorption ability is the chemical affinity of the analyte to the constituent layers. M. N. Ghazzal et. al have studied TiO2 / SiO2 multilayer system and have demonstrated that tuning the hydrophobic–hydrophilic balance of the silica surface modifies the affinity of the mesoporous multilayer system to the environment [ref. 28 in the revised manuscript.]. Similarly we may assume in our study that the presence of a small amount of silica increases the chemical affinity to acetone vapors thus enhancing adsorption processes.

Q4:“Couple of writings over the manuscript lack relative background literature/evidence, as shown below:

Page 2 ‘hard-template method also known as nanocasting where a solid template with well-defined porosity on nanometer scale is infiltrated with target material and after removal of the template a functional structure with inverse morphology is obtained.’

Page 9 ‘etching increases the porosity in the films thus facilitating the adsorption and condensation of vapors in the generated pores that finally leads to an increase in effective refractive index of the films.’

In accordance to the reviewer recommendation, in the revised manuscript we have added 6 additional references – refs. 20, 21, 22, 23, 24 and 26 and revised the manuscript accordingly.

 Q5: “It necessary to discuss more about the differences or dis/advantages of soft and hard-template method, since mentioned both of them earlier.

 In accordance to the reviewer recommendation, we have revised the introduction and stress the reader attention on pros and cons of soft and hard template and the difference between the two approach for preparation of porous materials. We have added 5 additional references – refs. 20, 21, 22, 23 and 24 and revised the manuscript accordingly.

Reviewer 2 Report

The manuscript “Hard-Templated Porous Niobia Films for Optical Sensing Applications" by Babeva et al. presents the preparation of porous Nb2O5 films obtained by a modified hard-template method and their optical and sensing application. The manuscript is appealing, well-written, and fits the journal’s scope. The film characterization is well done and the conclusions are in agreement with the experimental results. In my opinion, its content is well worth publishing in Photonics. However I have some questions and I will indicate some minor changes, which I believe could improve the scientific impact of the manuscript. In this way I have the following comments regarding the manuscript:

- Figure 1b lacks resolution. Please revise.

- Regarding the data presented in Figure 4, it is possible to compare similar systems in the literature? Same as the results presented in Figure 6.

- It is not clear in the manuscript the sensing properties highlighted in the abstract. After reading the manuscript title and the abstract I was induced to expect some sensing application in this investigation, which is not the case. I mean, it is not clearly presented in the manuscript. For instance, regarding the synthesized material, how did the authors “study their suitability as active media for VOC’s sensing” in this manuscript? (Page 2, lines 68-69). Please note that after a careful reading of the manuscript, I could get the answer. However, all sensing investigation is mixed with the material characterization. I strongly suggest to the authors to better present the sensing results to highlight these results among all the others.

Author Response

Dear Reviewer, thank you very much for your positive assessment of our work and for your fruitful recommendations and very helpful comments. Please find as follows our answers. Please have in mind that all your comments and recommendations are considered and the manuscript text is revised accordingly everywhere it is necessary. The changes in the text are in red color.

“Figure 1b lacks resolution. Please revise”

Answer: Figure 1b has been revised

“Regarding the data presented in Figure 4, is it possible to compare similar systems in the literature? Same as the results presented in Figure 6.”

Answer: It is very difficult to find similar data in the literature. Even it won’t be an exaggeration to say that it will be impossible. It is because reflectance is not an intrinsic property of the material but it depends on many parameters such as refractive index, extinction coefficient and thickness of the film, wavelength range, incident angle and polarization and substrate type. Therefore the probability to find the same combination of materials, the same ratio of both oxides, the same spectral range, the same etching time etc. is very low. The same is for the results presented in Figure 6.

“It is not clear in the manuscript the sensing properties highlighted in the abstract. After reading the manuscript title and the abstract I was induced to expect some sensing application in this investigation, which is not the case. I mean, it is not clearly presented in the manuscript. For instance, regarding the synthesized material, how did the authors “study their suitability as active media for VOC’s sensing” in this manuscript? (Page 2, lines 68-69). Please note that after a careful reading of the manuscript, I could get the answer. However, all sensing investigation is mixed with the material characterization. I strongly suggest to the authors to better present the sensing results to highlight these results among all the others.”

Answer: We agree with the reviewer and have revised section 3.0. “Results and discussions” by dividing it into four subsections. All sensing results and discussion could be found in subsection 3.4.